# Rethinking Urban Cat Management—Limitations and Unintended Consequences of Traditional Cat Management

**DOI:** 10.3390/ani15071005

**Published:** 2025-03-31

**Authors:** Jennifer Cotterell, Jacquie Rand, Rebekah Scotney

**Affiliations:** 1Australian Pet Welfare Foundation, Kenmore, QLD 4064, Australia; 2School of Veterinary Science, The University of Queensland, Gatton, QLD 4343, Australia; j.rand@uq.edu.au (J.R.); rebekah.scotney@uq.edu.au (R.S.)

**Keywords:** legislation, enforcement, free-roaming cats, community-based solutions, urban cat management, One Welfare, animal management officers, feral cats, domestic cats, semi-owned cats

## Abstract

Australia’s management of free-roaming cats has traditionally relied on legislative mandates requiring cat owners to confine, sterilize, register, and microchip their cats, with penalties for non-compliance. However, these enforcement-driven policies face significant challenges. They are costly for local governments, resource-intensive, and fail to address the root causes of free-roaming cat populations, such as financial barriers and the prevalence of semi-owned or stray cats, particularly in disadvantaged areas. Animal management officers are central to enforcing these measures, often issuing fines and trapping cats identified as causing a nuisance. Despite these efforts, compliance remains low, and issues like high shelter intakes, cat-related complaints, and euthanasia persist. Moreover, the punitive nature of these policies can place additional financial strain on vulnerable communities and negatively impact the mental health of animal management officers and shelter staff. This approach, focused on penalties, addresses symptoms rather than systemic issues. A shift toward addressing the root causes—through financial support, including support for cat sterilization, resource accessibility, and community engagement—presents a more effective and compassionate solution. Such strategies benefit both the cats and their caregivers while reducing the burden on local governments, promoting sustainable and humane outcomes for communities while better protecting wildlife.

## 1. Issues Associated with Free-Roaming Cats in Urban Areas

Free-roaming cats in urban and peri-urban areas are frequently associated with complaints from residents to local governments (councils). Complaints often relate to defecating and urinating, property damage and disturbances from noise, as well as perceived risk to wildlife from predation and disease risk to humans, other pets, and wildlife [1,2,3]. Most free-roaming cats in urban areas of Australia are unidentified, lacking both microchips and collars or tags, and are classified as “stray” in shelter data [4]. They are rarely reclaimed by an owner if impounded, with reclaim rates from shelters and local government animal facilities (pounds) in Australia averaging approximately 5% [4]. Although some of these are unidentified owned cats, many are not owned. Typically, these cats receive care from compassionate community members who provide regular food and other necessities. These caregivers, frequently referred to as semi-owners, do not perceive the cats as their property [5].

In Australia, 3–9% of adults feed one or more cats daily without claiming ownership, averaging about 1.5 cats per caregiver (semi-owner) [6,7]. Most of these cats are not sterilized, contributing significantly to the population of unwanted kittens. Additionally, many of these cats are poorly socialized to humans and tend to be timid or fearful, placing them at higher risk of euthanasia in shelters or pounds [8]. Free-roaming cats and shelter and pound admissions of cats are most prevalent in low socioeconomic areas, defined by the Australian Bureau of Statistics as regions with limited access to materials and social resources, and have a restricted ability to participate in society [9].

With recent increases in the cost of living, those in low socioeconomic areas face even further budgetary concerns, especially when having to choose between their pets’ needs or those of other family members [10]. This contributes to lower sterilization rates, especially in cats under two years of age, resulting in unplanned litters of kittens [10,11]. Pet care and veterinary costs, as well as costs for compliance with animal welfare and regulation policies (registration and containment laws), all add to the budgetary burden on low-income households. The ability of disadvantaged communities to abide by the legislative requirements, especially containing cats on their property, is much reduced [10,12]. Because many of these cats are not sterilized or fully contained, they are at greater risk of causing nuisance complaints associated with fighting, urine spraying, or yowling during the breeding season, which can result in impounding and possible euthanasia [13].

For cat semi-owners residing in disadvantaged areas, if provided with assistance, many are willing to assume full responsibility for the cats they care for, including microchipping under their name and registering (licensing) as the owner with the local council [13,14,15]. In instances where semi-owned cats are provided with regular food by a household or individual, transferring full responsibility for these cats to these caregivers and assisting them to have their cats sterilized effectively halts the breeding cycle. This approach facilitates a long-term outcome where the cat becomes owned by an individual responsible for their care, thereby contributing to improved welfare and population management. However, this can only occur if sterilizing, microchipping, and registration are affordable and the new owners are exempt from the enforcement of regulations related to excess cats and associated permits, and containment when this is not feasible or affordable. Their participation ultimately supports reducing stray populations and improving welfare outcomes. Despite this, most subsidized cat sterilization programs that are currently available remain financially out of reach for many low-income cat owners and semi-owners, and the enforcement of bylaws that cannot be complied with is still the norm, contributing to the ongoing maintenance of free-roaming urban cat populations and their associated issues.

## 2. Traditional Management of Free-Roaming Urban Cats

### 2.1. Legislative and Enforcement-Based Approach

In Australia, the traditional method for the prevention and management of free-roaming cats is twofold—legislation, followed by enforcement aimed at achieving compliance. Firstly, state legislation and local government bylaws (ordinances) are enacted and, depending on the state and local government area, require cats to be contained to the owner’s property, sterilized, registered, and identified with a microchip, and limit the number of cats allowed to be kept. In most states of Australia, including Victoria, local governments (councils) are responsible for domestic cat management and the enforcement of state government legislation. For example, under the Victorian Domestic Animal Act 1994 [16], cats over three months of age must be microchipped and annually registered with the council they reside in. Victorian state government legislation requires that pet cats not trespass on private property or cause a nuisance. In addition, approximately 42 out of the 79 Victorian local government areas have local bylaws that require cats to be contained within the owner’s property, with curfews either from dusk to dawn or 24/7, and a further 16 other local government areas are considering introducing cat curfews [17]. Despite the increasing adoption of cat containment bylaws, there is currently almost no peer-reviewed research evaluating their impact. These policies are primarily driven by public opinion and are implemented with the intention of reducing cat-related calls to councils and mitigating wildlife predation. However, the absence of evidence-based studies raises significant concerns about the potential social, environmental, and welfare implications of such measures, highlighting the urgent need for rigorous scientific research to measure effectiveness.

Secondly, in alignment with the legislative framework, enforcement and compliance activities are used to deal with free-roaming cats causing a nuisance and resulting in a complaint to the council. For example, in Victoria, under Section 23 of the Domestic Animal Act 1994 [16], in cases where an owner is identified, they can be served a “notice of objection”, issued by an authorized officer within 5 business days of the offense, objecting to their cat’s presence trespassing on private property. If their cat is trespassing, the “notice of objection” requires an owner to rectify the cat trespass problem within a specified timeframe or they will subsequently be issued a fine for non-compliance, with increased fines for subsequent offenses, with unpaid fines leading to a summons to attend court. Where an owner is not identified, animal management officers (AMOs) typically trap nuisance or wandering cats, or trap cages are provided to the public for this purpose. Cats without identification who are not reclaimed by an owner within the legislated holding period are then either rehomed or euthanized. The legislated holding period depends on the state and is typically between three (Queensland, QLD) and eight (Victoria) working days for unidentified cats [18].

However, cat management practices are not consistently undertaken by all local governments. For instance, in New South Wales (NSW), some councils do not proactively enforce legislative requirements. This includes a failure to record cat-related inquiries, which prevents the collection and comparison of relevant data. Additionally, some councils do not conduct trapping programs or accept stray cats, further limiting their involvement in cat management [19]. As a result, in these councils, the responsibility for bringing stray cats to the local government animal facility (pound) or shelter is left to the community. This lack of formal cat management gives the impression that the council does not encounter significant cat-related issues. Additionally, when enforcement actions are implemented, council officers require proof of ownership to pursue compliance or enforcement—a requirement that cannot be fulfilled in cases where there is no identifiable owner [19,20].

In many pounds and some shelters, cats appearing to be fearful or aggressive in a trap cage or shortly after admission are inappropriately considered feral; therefore, as feral pests, they can be immediately killed without undergoing the legislated hold period [21]. Definitions of feral and domestic cats in state government legislation vary from state to state and even between different legislative Acts in the same state, for example between legislation related to biosecurity and domestic animal management. It is strongly recommended that for the successful management of cats living around areas where humans live or frequent and thereby obtaining food intentionally or unintentionally, they should be considered domestic cats, as recommended by Australia’s peak animal welfare organization, the Royal Society for the Prevention of Cruelty to Animals (RSPCA). In their report on Identifying Best Practice Domestic Cat Management in Australia [22], they recommended that cats be classified as either feral or domestic, with domestic cats living in the vicinity of where humans live or work and obtaining food from humans intentionally (owned or semi-owned cats) or unintentionally (unowned cats) [22]. Because domestic cats live around humans, they are a cause of complaints associated with nuisance behaviors. Distinct from domestic cats, feral cats live and reproduce remotely from humans and are not a cause of complaints relating to nuisance behaviors. They are considered pest animals that can be poisoned, shot, or killed by blunt trauma [23,24]. If only owned cats are considered domestic cats, and all others are feral cats, this generally limits management to lethal methods, which have failed for decades to successfully manage these cats at town, city, and state levels. In contrast to Australia, in North America, the term feral is commonly used for cats that are receiving food intentionally or unintentionally from humans, and these would be considered semi-owned and unowned domestic cats by the RSPCA.

### 2.2. Role of Animal Management Officers (AMOs) in Urban Cat Management

In most states of Australia, the primary role of traditional AMOs (in the USA, animal control officers (ACOs) is to provide community safety by promoting responsible pet ownership and enforcing domestic animal legislation. The job role focuses on community safety by resolving animal-related complaints, educating the community, impounding stray and wandering dogs and cats, investigating dog attacks, and enforcing relevant laws. The first author (J.C.) has 22 years of experience in discharging these responsibilities, including in leadership roles [25,26]. Enforcement actions can range from issuing fines to pursuing charges and summons in a magistrate’s court for more serious offenses or unresolved, persistent complaints. This cost is borne by the council and typically exceeds the income received from fines and penalties [21]. Animal management officers often note that issuing infringement notices and fines to low-income cat owners and semi-owners does not lead to compliance, especially when residents cannot afford the costs of containment systems or sterilization, nor do residents see it as a priority when the greater importance for household income is the provision of food, housing, power, and transport. Historically, AMOs have repeatedly visited the same properties to address recurring nuisance cat issues and unregistered cats, often by issuing infringements when an owner could be identified. If infringements are not paid by the accused, councils are required to allocate additional financial resources to pursue further action, including potential court proceedings to recover outstanding payments. This process may result in the establishment of payment plans, which in turn require ongoing resource commitments for monitoring and enforcement. Alternatively, the cat is surrendered or is trapped and impounded, resulting in further costs to the council if it is not reclaimed.

Because the primary role of AMOs is ensuring community safety and enforcing domestic animal legislation, it has contributed to a negative public perception of their presence. AMOs are often associated with responding to complaints or addressing animal welfare issues, rather than with providing support for the cat caregiver, their cats, and the complainant, and providing practical information and assistance to resolve the issues [27]. This perception is particularly pronounced in low socioeconomic areas, where a general distrust of authority exists [28]. Animal management officers are frequently viewed as “fine issuers” rather than as facilitators of welfare or compliance. This dynamic further exacerbates challenges in fostering constructive relationships between AMOs and these communities. The compliance-driven approach to addressing nuisance cat complaints typically involves trapping and impounding cats, which is associated with low numbers of cats being reclaimed by owners and high euthanasia rates. This contributes to the community’s perception that the primary role of an AMO is to trap cats, with a predictable outcome—euthanasia [27].

### 2.3. Working in Silos

In many states, including Victoria, animal welfare agencies, AMOs, and rescue groups often operate in silos, with limited collaboration or coordination. This fragmented approach undermines efforts to address complex issues like free-roaming cat populations and overcrowded shelters and pounds. Each group tends to focus on its specific mandate—rescue groups prioritize rehoming, welfare agencies operate shelters that accept owner-surrendered cats and stray animals, particularly if they are sick or injured, and may also have a contract with the local government to take impounded animals, while AMOs concentrate on enforcing regulations such as containment and registration laws. The lack of communication and shared strategies between these entities can lead to inefficiencies and missed opportunities for systemic solutions.

One obvious example is the handling of semi-owned or unowned cats in low-income communities. While rescue groups may attempt to rehome cats, they often lack the resources or authority to address the root causes, such as limited access to affordable cat sterilization and veterinary services. Simultaneously, AMOs enforcing punitive measures such as trapping and fines may inadvertently alienate the community, creating resistance to engagement. Meanwhile, welfare agencies may advocate for policy changes but often have insufficient resources to implement large-scale interventions. This disjointed approach can result in high euthanasia rates, persistent cat-related complaints, financial strain on councils and negative impact on staff. This demonstrates the urgent need for collaborative frameworks that pool resources and expertise to deliver more effective and humane outcomes.

### 2.4. Human and Animal Welfare Consequences of a Legislative and Enforcement-Based Approach

The outcomes of this compliance-based approach are that excessive numbers of cats are impounded nationally, exceeding the capacity to rehome them, resulting in approximately 46% of cats entering local government facilities (pounds) in Australia being euthanized [4]. The worst-performing quartile of local governments impounding more than 50 cats in a year euthanized between 67% and 100%. Because many shelter and pound staff are required to regularly kill healthy and treatable cats and kittens, it negatively impacts their job satisfaction and mental health. This increases their risk of depression, traumatic stress, substance abuse, high blood pressure, sleeplessness, and suicide [29,30,31,32,33,34]. This legislative and enforcement-based approach has not substantially reduced cat-related complaints or impoundments because it does not address the underlying causes [4]. Understanding the underlying contributing factors to free-roaming cats is critical for understanding why each of the typical legislative approaches is ineffective.

## 3. Legislative Approaches: Limitations and Unintended Consequences

### 3.1. Mandated Containment of Cats to the Owner’s Property

Most states and territories in Australia either have state- or local-government-level mandated containment (leash laws in the USA) or are currently considering their introduction [35,36,37,38]. These mandates are based on a lack of understanding of the causes of free-roaming cats, and a mistaken belief that it is irresponsible cat owners who are the cause of the problem, whereas, in reality, most free-roaming cats have no identifiable owners and are in low socioeconomic areas. Mandates for cat containment that are targeted at cat owners cannot be enforced when there is no identifiable owner of a stray cat [13].

A common approach used by councils for implementing new local laws, such as mandatory cat containment, involves conducting online surveys. This method is relatively low in cost and requires significantly less time and fewer resources, compared to hiring a third party to undertake consultations with stakeholders and experts in domestic cat management to produce a comprehensive report. The online surveys typically include straightforward questions, such as “Would you support the introduction of a cat curfew?” Although discussion papers are occasionally produced to provide additional context on the issue, these typically focus on the risks free-roaming cats pose to wildlife. Rarely do they offer residents information on the underlying drivers of free-roaming cat populations or evidence-based solutions to address the issue. When community consultations yield a predominantly affirmative response, the new legislation is typically implemented on the basis that it reflects public consensus [39,40].

Cat containment mandates are not successful in the short or long term because these laws do not relate to semi-owned and unowned cats and, in fact, lead to increased cat-related complaints, potentially because residents have an expectation that they will not see free-roaming cats [41]. For example, the Shire of Yarra Ranges mandated a 24/7 cat curfew in 2014; three years later, cat-related nuisance calls had increased by 143%, cat intake was 68% higher, and the number of cats euthanized was 18% higher than at baseline (2014), even though the human population had only increased by 2% in that time [41]. The city of Casey’s council introduced a 24/7 mandatory cat curfew in 1998. However, 20 years later, cat intake was still 296% higher than at baseline, while the human population had only increased by 134% [41].

Victorian councils acknowledge that mandating confinement is unenforceable, with minimal income being generated from fines associated with curfew breaches [42,43]. There is no well-designed research to show whether these mandates have any impact on reducing cat-related complaints or free-roaming cats, including stray and semi-owned populations. In addition, there is no research from Australia documenting whether this legislation is being enforced and, if so, an analysis of the outcomes. The ‘RSPCA Australia: Identifying Best Practice Domestic Cat Management in Australia 2018’ report acknowledges: “Overall, councils with cat containment regulations have not been able to demonstrate any measurable reduction in cat complaints or cats wandering at large following the introduction of the regulations”. The introduction of confinement mandates gives communities an unrealistic expectation that these laws can, and will be, enforced, with few or no further resources provided to undertake enforcement [42]. In the USA, it has been recognized by numerous welfare agencies and local governments that cat confinement laws are unenforceable in the absence of an identifiable owner, and have been rescinded [35,44,45,46].

In reality, most free-roaming cats and cat-related complaints emanate from low socioeconomic areas, where a higher proportion of residents live in rental properties, which often lack screens on windows or doors and rarely have cat-proof fencing. The cost of installing cat-proof fencing or containment systems typically ranges from A$700 to A$2000 or more and poses a significant barrier, especially since tenancy agreements often prohibit property modifications. Three Victorian councils recently rescinded motions for 24-h cat curfews [47,48,49], with the reasons given being related to the impact on disadvantaged residents and tenants. For example, one councilor (elected official) stated that “the financial cost burden the policy would have imposed upon residents on low, fixed incomes may have required them to give up their cat, which in many cases may be their only companion. That’s not something I could support, particularly in a cost-of-living crisis” [48]. Another council that rejected the motion for 24/7 cat containment issued a statement that “this decision recognizes the challenges that 24-h cat containment would impose, particularly for older and low-income resident cat owners, as well as renters who may not be able to adapt their home to ensure their cat could not escape”. It was recognized that mandating containment could lead to the relinquishment of companion cats, particularly with the current cost of living crisis worsening the affordability of cat containment systems [48].

Mandated cat containment does not address the stray or semi-owned cat population. The city of Melbourne recently reported that most of the cat-related complaints they receive are for semi-owned or stray cats, while 74% of cats registered by owners are confined in apartments and 97% are sterilized [50]. An additional issue is that enforcement officers may mistakenly impose fines and penalties intended for pet cat owners on the caregivers of semi-owned and stray cats. The misapplication of domestic animal legislation to semi-owners is ineffective and costly, especially if this is disputed in a magistrate’s court process because, when a responsible owner cannot be definitively linked to the cat, the charge cannot be proven and is dismissed, but the council still incurs the costs. Enforcement measures and legislative requirements designed for identified pet cat owners cannot be applied to semi-owners or caregivers who have a stray cat that comes onto their property [46].

Moreover, containment measures do not guarantee that “door-dasher” cats or those animals that are difficult to contain will remain confined. Notably, 41% of lost cats were described by their owners as indoor-only, but escaped through open windows, doors, or garages [51]. Another study reported that of those cats who went missing and had been indoor-only during the previous six months, 22% had previously escaped at least once before going missing [52], and a recent Australian study reported that 5% of cats contained 24/7 inside the owner’s property had inadvertently escaped in the previous two weeks [11].

### 3.2. Mandated Sterilization

Mandated sterilization is ineffective for the same reasons. It is rarely enforceable and fails to recognize that it is not a lack of motivation but, in most cases, a lack of money that creates a barrier for owners to sterilize their cats. In fact, family income is the strongest predictor of whether a cat is sterilized [53,54].

Sterilizing a cat before it is sold or given away is mandatory in Western Australia, South Australia, and ACT, but not in Queensland, New South Wales, and Victoria. A national survey in 2022 found that the proportion of cat owners with sterilized cats was not different between jurisdictions with and without these mandates [55]. In addition, a recent study reported that the only three states with mandated sterilization in Australia also had the highest per capita cat intakes into shelters and pounds, indicating that mandated sterilization is not effective in reducing free-roaming cats and related issues [4]. In a 2016 study of 191,000 cats entering RSPCA facilities nationally, the ACT was the only state or territory with mandated sterilization (by six months of age) but it recorded the second-highest proportion of unsterilized cats under six months among admissions. This finding further highlights the limited effectiveness of mandated sterilization policies [56]. The ACT enacted mandatory sterilization for cats in 2001. In 2000, cat intake into the ACT RSPCA shelter was 7.69 cats/1000 residents and by 2006, cat intake was 8.02 cats/1000 residents [57]. This suggests that the sterilization mandate made no positive impact on the shelter intake of cats, likely because legislators failed to recognize the underlying causes, which are cost and accessibility barriers for low-income cat owners, and that most shelter admissions are of unidentified owned, semi-owned, or unowned domestic cats [14].

### 3.3. Mandated Registration

Most states of Australia require cats to be registered (licensed) with the local government and to be microchipped or, in some cases, wear an identification tag. This legislation is aimed at owned cats, where a designated individual, classified as the “owner” for legal purposes, can be held accountable for enforcement. Again, this legislation fails to address the issue of the semi-owned or unowned cats present in communities. These animals do not have a designated owner, rendering enforcement ineffective in these cases. According to data gathered over three financial years (up to June 2023) from the Sentencing Council in Victoria, there were just 3506 recorded charges for failure to register a dog or cat, despite there being an estimated 1.7 million pet dogs and 1.5 million pet cats in Victoria. Of these, 85% of cases resulted in further fines being issued; 12% were adjourned, with conditions to register the animal, 3% were dismissed, and 0.03% resulted in community correction orders [58]. These figures, although not distinguishing between dogs and cats, underscore the financial burden associated with recovering the initial infringement fees and the relatively low levels of prosecution for non-compliance, given that there are over 3 million pet dogs and cats in Victoria [55]. A survey conducted with councils across Australia revealed that two-thirds required cat registration. However, an analysis of the data provided by these councils estimated that only one-third of pet cats were actually registered [59].

Local councils face further costs related to administrative tasks associated with the enforcement of registration, including issuing demand letters, preparing summonses, providing legal representation for magistrate court appearances, and allocating officer time for testimony. These cumulative expenses can total A$1500 to A$3000 or more to initially recover a penalty of approximately A$350, with costs rising significantly if the accused contests the original fine. Moreover, this enforcement process may fail to ensure registration, with continued non-compliance likely leading to further fines, exacerbating the issue, and may lead to relinquishment of the cat, further increasing council costs. In Queensland, the mandatory registration of cats was repealed in 2013 because this was “considered to deliver the greatest net benefit to stakeholders, as it yields the greatest potential red tape reduction, and cost savings to local governments and cat owners, without compromising reunification outcomes and euthanasia rates” [14,60].

In NSW, registration and breeder permit fees present significant financial barriers to cat ownership. When acquiring a domestic cat more than four months old that is not sterilized, owners are required to pay an annual breeder permit fee of A$96, in addition to a lifetime registration fee of A$69. If the cat is more than four months old, the breeder permit fee for 1 year remains applicable, even if the cat is promptly sterilized, microchipped, and registered at or shortly after the time of acquisition [61,62]. The outcome is that return-to-owner rates in NSW are considerably lower compared to Victoria and Queensland, standing at 3% versus 7%, respectively [4]. This discrepancy may be partly attributed to the use of the state microchip register, which can identify cat owners who have not paid registration fees, thereby creating a financial disincentive to microchip their pets.

Excluding the administration costs to councils in NSW for managing cat registration, council costs for managing cats is estimated to be 7 to 10 times greater than the revenue generated for the state government from registration fees [62,63]. This suggests that imposing financial barriers, which discourage owners from microchipping, is fiscally inefficient and undermines return-to-owner rates. A more effective approach would involve abolishing mandatory registration and focusing on making microchipping more accessible, ideally through integration with free or low-cost sterilization services. The utility of microchips in reuniting cats with their owners would be enhanced through regular reminders via SMS or email, prompting owners to update their contact information when needed [63,64]. Additionally, breeder permits for cats over four months of age should be discontinued, along with registration, because these requirements increase the financial burden on owners and hinder participation in sterilization and microchipping programs, or alternatively, increase the cost to organizations including these in sterilization and microchipping programs.

In Victoria, the legislation mandates that a source number must be obtained before implanting a microchip into a dog or cat born after 1 July 2020 [65]. This requirement, intended to promote ethical pet breeding and enhance the traceability of dogs and cats, applies to ownership transfers, including taking ownership of a stray cat, as well as advertisements for sale or giving away. Specifically, advertisements must include both the microchip number and a source number generated through the Pet Exchange Register (PER), which incurs an additional cost of A$24.50 [65]. Non-compliance with these regulations constitutes an offense, carrying a maximum penalty of A$987 for individuals and A$3951 for the publishers of non-compliant advertisements. However, the introduction of the Pet Exchange Register has had unintended consequences. Semi-owners of cats—individuals informally caring for and feeding unowned cats—face barriers to taking legal ownership. These individuals are now unable to access veterinary services for sterilization and microchipping without first obtaining a source number. Similarly, carers of community cats are often turned away from veterinary clinics due to the lack of a source number. Furthermore, as the Pet Exchange Register is accessible only online, it limits access for socioeconomically disadvantaged individuals, exacerbating the challenges faced by these communities. It is recommended that the requirement for a source number for cats is abolished, given that it is a barrier to the effective management of urban cats and that purpose-bred cats constitute a small percentage of pet cats, particularly in low socioeconomic areas [15,55].

Furthermore, councils that do not operate their own pound facilities and, therefore, do not engage in rehoming animals may lack awareness of the requirement to obtain source numbers to conduct cat sterilization programs. This oversight could stem from a lack of familiarity with regulations primarily designed for breeders, sellers, and rehoming organizations, who already have a source number for their business. Additionally, these councils may not fully understand the administrative implications of the Pet Exchange Register, as their focus is often directed toward broader community management and animal control efforts. An example of the impact of this gap in awareness is where councils provide residents with vouchers for free or low-cost cat sterilization and microchipping, but veterinarians cannot microchip the cat without a source number. This hinders compliance with legislation as proof of microchipping is needed to register the cat. Therefore, the requirement for a source number is a barrier to councils addressing the needs of disadvantaged owners and semi-owners within the local government area.

### 3.4. Cat Limits and Excess Cat Permits

Limits on the number of cats per household, typically set at 2–4 cats [66], along with the high permit fees required to exceed these limits, may discourage semi-owners from fully adopting stray cats. This is particularly relevant for individuals who already have companion cats. Currently, no scientific evidence supports a link between the number of cats in a household and an increase in nuisance complaints or public health risks [14]. In fact, a single neglectful owner could cause more complaints than a conscientious owner with ten cats. Policies should prioritize addressing the specific impacts of individual cats rather than enforcing arbitrary household limits. Existing regulations that address nuisances and public health hazards are adequate, rendering household cat limits unnecessary. Moreover, these restrictions and associated fees are unlikely to prevent residents from caring for additional cats but may deter them from officially adopting or sterilizing them. A more effective strategy for managing urban cat populations is eliminating household cat limits and associated costly permits and instead, implementing an assistive approach to resolve cat-related complaints, and where necessary, utilizing existing animal welfare and anti-nuisance laws.

To effectively manage and reduce problematic cat populations in the community, where there are multiple cats at a site, provision of affordable sterilization and microchipping for owners’ and carers’ cats, and encouraging semi-owners to take ownership of the cats they are caring for is a more practical and sustainable solution. Providing these resources can encourage caregivers to seek assistance without fear of their cats being impounded or penalties levied for excess cats, because trust is essential for their effective management. Addressing these challenges proactively can help mitigate cat population growth and related nuisances before they occur.

### 3.5. Feeding Bans

In many jurisdictions, it is illegal to feed a pest animal without a permit, except with the intention of trapping and killing it [67,68,69]. In some states, this extends to feeding stray cats. For example under the Queensland Biosecurity Act of 2014, only owned cats are considered domestic cats and it is illegal to feed, move, adopt, or sell a cat that is not owned, because they are considered “restricted matter” [67]. The Brisbane City Council (BCC) acted “to address community concerns and prevent further feeding and release of strays” and “carried out a program of compliance and enforcement action to improve their legislated biosecurity risk management” [21]. This resulted in 52 convictions being handed down for five individuals, totaling A$27,000 (A$ 519.23 per conviction), and a three-month imprisonment term for an elderly lady [21]. The multiple convictions per person highlight the challenge of an enforcement approach because people will continue to feed cats for reasons including: “I feel sorry for them”, “There was just nobody there to help”, ”I feel responsible”, ”I just really wanted to see these cats taken care of”, and, because of their relationship with the cats, “They’re basically the same as a pet cat that you’d have at home, they have names, they have personalities”, “They are the reason I get up”, “They’re my babies, sort of like they’re my kids”, and “I love them. I love them” [12,15,70,71,72].

Considerable expense is incurred when issuing fines and summons for court appearances, which typically far exceeds income. Although they can be effective at a limited number of individual sites when cats are simultaneously trapped and removed by the authorities, such activities are extremely resource-costly when legal costs for court appearances and trapping costs are considered, and ignore the human impact of this approach. Trapping costs were reported at A$1.3 million over 5 years, requiring 9.4 trap nights per cat at a cost of A$241.41 per cat [21]. Although effective at a colony level, albeit costly, this approach is not effective in a suburb or at city level because of the number of individuals feeding stray cats. For example, at one site in Brisbane, 52 charges were laid for 5 individuals; however, an estimated 60,000 people (3% of adults) in the city of Brisbane’s population of approximately 2.5 million were feeding cats, and they each fed an average of 1.5 cats (90,000 cats) [6]. Rather than the lethal management that is prescribed by legislation which views semi-owned cats as feral pests, it is strongly advised that the RSPCA’s recommended classification for feral and domestic cats is adopted to facilitate effective urban cat management that is based on One Welfare principles [73].

The comparative cost and feasibility of trapping and enforcement compared to a program based on sterilizing cats depends a great deal on who is funding and carrying out the trapping and funding the sterilization costs. Where the intention is a trap–adopt or kill program, most or all of the costs are paid by local governments, including trapping, impoundment, and euthanasia costs. In contrast, in a non-lethal program based on sterilization, welfare agencies, rescue, and other groups typically utlize volunteers, or, where feasible, semi-owners for trapping and transporting cats and often fund sterilization costs, thereby reducing costs to local governments for managing cats. Therefore, cost comparisons vary significantly based on whether labor is voluntary or paid, the cost of sterilization, and the relative cost of sterilization versus euthanasia.

### 3.6. Trap-Neuter-Return (TNR) and Return-to-Field (RTF, Shelter-Neuter-Return)

In Australia, practices such as sterilizing free-roaming cats that are being cared for by caregivers (semi-owners), often termed trap-neuter-return (TNR), are prohibited under biosecurity, containment, and abandonment legislation, though the latter has not been legally challenged. Despite these restrictions, TNR programs are covertly implemented in urban areas throughout the country and have been shown to reduce cat populations by 30% at targeted sites within two years [74,75,76]. Similarly, return-to-field (RTF)—where poorly socialized but healthy cats that are unlikely to be adopted are sterilized and returned to their original locations, under the assumption they are being fed—remains illegal.

Both TNR and RTF are vital for mitigating the psychological strain experienced by shelter staff and community members involved in their care. The process of trapping and transporting cats for euthanasia has significant emotional repercussions for AMOs, shelter workers, and the carers of the euthanized cats [13,29,31,33,70,77,78]. Therefore, further consideration is warranted regarding the documented mental health toll on individuals who care deeply for animals, including carers feeding stray cats, shelter staff tasked with euthanizing otherwise healthy or treatable animals, and AMOs repeatedly trapping cats at multi-cat sites and transporting them for euthanasia [13]. In shelters committed to minimizing euthanasia, timid or fearful stray cats may be confined for extended periods in attempts to socialize and rehome them. When these efforts fail to get them adopted, the euthanasia of these cats can have a profound negative emotional impact on the shelter staff who were tasked with interacting with them and gaining their trust. This cumulative stress, including an increased risk of suicide, underscores the need to urgently address legislative barriers to RTF.

A program in Queensland based on sterilizing owned and semi-owned cats and an assistive approach to prevent cat surrender and impoundment (termed a Community Cat Program) demonstrated the effectiveness of this comprehensive approach. Owned and semi-owned cats were sterilized, and where possible, semi-owned cats were adopted by their carer. Poorly socialized impounded cats that were not reclaimed, were returned to where they were found rather than being euthanized. Over 3.4 years, the sterilization of 94 cats per 1000 residents resulted in a 60% reduction in shelter intake, an 85% decline in euthanasia, and a 39% drop in cat-related complaints in the third year [14]. The majority of sterilized cats comprised owned (74%) or semi-owned cats that became fully owned (10%), while 11% remained semi-owned at multi-cat sites, cared for by semi-owners. A small percentage (0.6%) were returned to field. These activities were conducted under a research permit issued by the Queensland Department of Agriculture and Fisheries (now the Department of Primary Industries) [79]. Notably, unowned domestic cats, defined as those surviving solely on unintended human food sources (e.g., waste bins), were not identified, presumably because compassionate individuals begin feeding these cats. However, under the research permit in Queensland, if a site with multiple unowned cats was identified, the intent was to sterilize and microchip them, and identify a carer to take responsibility for feeding and monitoring their health, so they were transformed into semi-owned cats.

Both sterilizing and returning semi-owned cats to their carers (TNR), as well as RTF (returning unclaimed cats to where they were found), reduce the mental health impacts of euthanizing healthy but fearful cats. However, in the Queensland Community Cat Program they comprised a relatively small proportion of the cats sterilized (12%) and at the level performed, would not be expected to substantially reduce the intake of cats. Nevertheless, these strategies are essential for reducing the psychological burden on shelter staff and other stakeholders and have been embraced by shelter staff. In the Queensland program, the primary value of Return-to-Field (RTF) lay in preventing the euthanasia of healthy cats that shelter staff had invested significant effort into socializing. These cats often exhibited some level of trust toward staff but not enough to be deemed adoptable within the available timeframe. The euthanasia of such cats was frequently a source of considerable distress for staff.

Community Cat Programs in Victoria, NSW and Queensland targeting owned and semi-owned cats have achieved similar results for decreases in intake, euthanasia, and cat-related complaint calls to those reported internationally for high-intensity TNR and RTF programs [13,14,75,80,81,82,83,84,85,86]. Only the Queensland program included TNR and RTF. However, both TNR and RTF are vital for protecting the mental health of staff in shelters and pounds, as well as community members. Allowing these practices would support both human mental health and animal welfare, and should be legalized, particularly in areas without species of conservation concern or where scientific evidence does not demonstrate harm to wildlife populations. While cats are often accused of causing extensive harm to native wildlife, population studies in urban, peri-urban, and small rural areas have yet to demonstrate a measurable impact on native bird and mammal populations. Habitat loss, rather than cat predation, has been identified as the predominant threat to these species [87,88,89].

Legislative amendments at the federal, state and local levels are urgently needed to enable the inclusion of TNR and RTF in sterilization initiatives, thereby mitigating the associated mental health impacts for those working in animal care who are exposed to the euthanasia of healthy cats and kittens. Sterilizing semi-owned cats at multi-cat sites (TNR) and providing care for them will not only reduce the negative mental health impacts of traditional management but also reduce the number of free-roaming unsterilized cats, thereby reducing nuisance issues and wildlife predation.

## 4. System Failures—Why a Better Way Is Needed?

The persisting challenges with free-roaming cats demonstrate that legislative and compliance-based approaches are not effective where authorities typically issue enforcement notices on residents to confine, sterilize, and register their cats (reactive approach) without thought or understanding of the reasons for non-compliance. This method of cat management prioritizes penalties for non-compliance over understanding and addressing community or individual needs and barriers to compliance. Enforcement-focused strategies also fail to acknowledge the interconnected relationships between people, animals, and their environments. This inevitably leads to perpetuating problems that do not benefit animal or human welfare. Furthermore, these strategies have not demonstrated a reduction in cat cat-related complaints or impoundments and, therefore, can only be viewed as continually failing strategies, despite considerable expenditure [4,63].

Contrastingly, the Community Cat Program implemented in the city of Banyule, Victoria based on an assistive approach providing free cat sterilization and targeted to suburbs and locations with high cat impoundments, reduced cat-related complaints by 36% and impoundments by 66% city-wide [13]. Other examples include a rural town with a population of 3000, located within the City of Ipswich, Queensland. The implementation of a Community Cat Program led to a 60% reduction in cat admissions to shelters and an 85% decrease in the number of cats euthanized in the third year of the initiative. Additionally, cat-related calls to local authorities dropped by 39% [14], consistent with unpublished camera-trap data demonstrating a significant decrease in numbers of free-roaming cats over time. Similarly, in the City of Parramatta, New South Wales, which is part of Greater Sydney and has a population of approximately 256,000, a program funded by the NSW Government in collaboration with RSPCA NSW resulted in a 46% decline in the intake of cats and kittens at the RSPCA Sydney shelter and a 41% decrease in cats impounded by the council. There was also a 49% reduction in cat-related nuisance complaints to the council [85]. In two smaller regional towns—Weddin (population 3608) and Walgett (population 5250)—cat-related nuisance complaints to the council decreased by 66% and 91%, respectively. These findings highlight the significant impact of Community Cat Programs in reducing the number of unwanted litters, roaming cats, and the associated issues by assisting cat owners and semi-owners to have the cats they are caring for sterilized and microchipped, and, where possible, to take ownership of semi-owned cats [13,14,85]. The success of these programs contrasts with the results of the traditional ‘trap and adopt or kill’ approach for urban cat management [4,14].

These examples demonstrate that successful cat management is achieved by taking a One Welfare approach, which recognizes the interconnectedness of animal welfare, human well-being, and their social and physical environments, and combines this with financial and resource support for disadvantaged owners and semi-owners. Acknowledging and supporting the strong attachment documented between semi-owners and the cats they care for [90], and implementing assistive management, results in positive impacts on human caregivers’ quality of life and the cats’ welfare [90]. Conversely, enforcement-centered approaches to cat management have negative impacts on human well-being and cat welfare and have not been demonstrated to positively impact urban wildlife populations [5,70,91].

## 5. Conclusions

Australia’s legislative and compliance-based strategies have been used over decades to manage free-roaming urban cats, but they have demonstrated significant limitations and unintended consequences. These approaches, which prioritize enforcement and penalties, fail to address the root causes of non-compliance, such as socioeconomic barriers, lack of access to affordable cat sterilization services, and the prevalence of semi-owned or stray cats. Enforcement disproportionately burdens vulnerable populations, strains local government resources, and contributes to persistently high shelter and pound intakes and euthanasia. The mental health impacts on animal care workers, including AMOs, shelter staff, and rescue volunteers, further highlight the need for reform. Frequent exposure to the euthanasia of healthy and treatable cats, compounded by the frustration of ineffective enforcement measures, contributes to depression, compassion fatigue, traumatic stress, and even suicide among these workers [29,30,31,32,33,78,92,93]. This underscores the human cost of current management strategies and the urgent need for a more compassionate and effective approach based on One Welfare principles. Fragmentation among animal welfare agencies, AMOs, and rescue groups also hinders progress, with siloed efforts failing to create systemic solutions.

Australian legislation requires employers to adopt a proactive stance in safeguarding their employees’ physical and mental health, ensuring that workplaces are both physically and psychologically safe [94,95]. The onus is on employers to implement practices that protect the mental health of staff. A One Welfare approach to urban cat management offers a sustainable and humane alternative by recognizing the interconnectedness of animal welfare, human well-being, and their physical and social environments. By integrating financial and resource support, community engagement, and evidence-based policies, this model of a Community Cat Program for urban cat management not only improves outcomes for cats but also protects the mental health of animal care workers, strengthens the communities they serve, and protects wildlife by decreasing the numbers of free-roaming cats.

## Data Availability

Relevant data are reproduced in the text.

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
