# Peer review of "Rethinking Urban Cat Management—Limitations and Unintended Consequences of Traditional Cat Management"

_animals, 2025, doi:10.3390/ani15071005_

Round 1
Reviewer 1 Report
Comments and Suggestions for Authors
This is a fascinating look at the underpinnings of a success cat management/community engagement program in a city in a suburb of Melbourne. I have a few very minor specific comments below. I wasn’t really sure what I would find in the manuscript based on the abstract/summary and introduction. Therefore, my primary recommendation is to make much clearer the purpose of this manuscript, summarize the key take aways, and early on illustrate that the program has another publication. I’ve suggested a reorganization which may address this (next paragraph). Also then edit slightly the abstract and summary to better highlight what made it successful including the on the ground engagement of the AMO which was critical.
I wonder if a different sequence would also flow better. Section 1 is foundational and about the traditional management in Australia, its limitations and unintended consequences. Section 3 is one paragraph and seems as it if should be moved into a transition location between section 1 and 2 (and rolled into section 1). Then section 2, is the Banyule program including that the details have been published elsewhere. This should also include the information about the magnitude of the program, numbers of surgeries and number of staff so that it is clear that this isn’t 100s of surgeries but was microtargeted to the important locations—that makes it more feasible. Then section 5 about what the benefits are would fit better after the program description and before the reasons the program was a success. Then section 4 is the final section before the conclusion about why the program succeeded and what readers should take with them. That way, the manuscript starts with the current problem, shares a successful solution, and finally lays out why that solution was so successful and what others should consider in designing such programs.
Title: Suggestion that instead of “Rethinking urban cat management” could this phrase more clearly link to an already completed program that when it was discontinued led to things going to back to the way they were before? So, something more like “An assistive-approach to urban cat management”. Just a thought.
Key word: community engagement, animal shelter
Line 54: “most are unidentified” that was just stated. Is this unowned instead?
Lines 64-7: please reference this sentence or edit.
Line 75: please reference this statement or edit. And what does “take full ownership” mean in this context with the laws and regulations?
Line 82-3: free services are subsidized; someone pays the cost. Please edit this sentence for clarity about what was being eliminated and what was available instead.
Line 105-9: is this a personal communication? Please indicate where this information came from. Similar comment for lines 125-6. If this is personal experience, just modify the sentences a bit to indicate that. Can you also define the full role of the AMOs? Clearly it is traditionally enforcement, but do they have other duties even in traditional settings?
Line 129-30: I think the section numbers aren’t quite right.
Line 133 and following: again, reference please. This is a really important point.
Line 148: please clarify that this is the human population that increased. And that “some Victorian councils”…
Line 166-7: the US situation sentence. Please reference this.
Lines 194-5: what was included in responsible cat ownership? This varies widely. Please add.
Lines 226-7: is this outlay what the program cost? Or what was saved? Please clarify in the text.
Line 270: where does 5 cats/1000 residents come from? And is it likely specific to this area of Victoria?
Line 287: please add the missing bit.
Lines 338-9: sentence needs an edit.
Line 447 typo.
Line 484: and to institutionalize in some way the program. It can’t be dependent on just a couple of people.
Line 544: is this paper online?
Section 4.7 tie to cost up above on page 4 (or the opposite).
Paragraph starting on line 558: please reference these statement. Hurley & Levy 2022 Frontiers in Vet Sci is one of the options.
Line 586: what does it mean to accept ownership? What are the advantages and disadvantages for the person? Lines 593-4: this bond is still possible and, in many cases, present without officially owning the cat. Please edit so that this argument is fully developed.
The fact that this was driven by the AMOs was critical!
References: please check the journal requirements, especially for capitalization of words in the title (e.g., 4 vs 3) and edit. Also check authorship formats.
References 13 & 14 appear to be the same, just accessed on different dates?
15: what is the website for this reference?
Cat laws cost what? Chip etc. elitist assumption of ability to pay.
Acknowledgements: perhaps the other AMO during this time? The service providers and community members who engaged?
Author Response
"Please see the attachment"

Reviewer 2 Report
Comments and Suggestions for Authors
Management of unowned cats is a problem globally, with municipalities struggling to find successful, cost-effective strategies that enjoy support from local communities. While this paper offers reflections on an innovative approach implemented in the local jurisdiction of Banyule, Victoria, it also left me with more questions than answers. While I sympathise with the authors’ frustration at the cessation of what, on the statistics provided, was a successful program, those questions should be answered for the paper to be publishable.
Essential points to address
1. There is substantial overlap between the content of this paper and the recent publication Cotterell, J.L.; Rand, J.; Barnes, T.S.; Scotney, R. Impact of a Local Government Funded Free Cat Sterilization Program for Owned and Semi-Owned Cats. Animals 2024, 14, 1615. https://doi.org/10.3390/ani14111615. While the paper under review rightly refers to the earlier publication, the overlap left me wondering what significant extra points are being raised to justify a further paper. Perhaps they need to be organised around the reasons for the cancellation of the initial program and how successful work can be continued in the face of such pressures.
2. I found myself wanting to know more about why the initial program (highly successful according to the data provided) was cancelled. The changes are described in lines 480-492, but the reasons for the cancellation are given as changes in staffing and lack of staff training. Given the cost-effectiveness of the program as described, why would the council retreat to paying more for less results? I felt that there must be more to the decision-making and wanted to see the full case. If the full case was made, there would be opportunity to structure the paper as a point by point evidence-based rebuttal.
Other points
1. The simple summary makes no reference to the program being discontinued. One could, on the basis of reading the simple summary alone, conclude that the program is still running.
2. The abstract does state that the program was discontinued but gives no reasons for this decision. In my opinion, those reasons, together with a rebuttal, should be the focus of the paper to distinguish it from Cotterell et al. 2024.
3. Lines 84-87 – I wondered how veterinarians might respond to subsidised desexing of cats. Is it a threat to business, or an opportunity for extra work?
4. Lines 131-138 – I was uncomfortable with this paragraph, which was all assertion and no evidence. Whether it was intended or not, the lack of reference to any particular council is probably an advantage here, because I could imagine an action for defamation if one was named. I’ve attached two reports from two different Australian jurisdictions (they are combined into a single file) indicating a far more thorough, evidence-based approach to consultation and evaluation of cat by-law programs. Are they representative? That I can’t say. They do, though, show that the assertions in the paper need support or they should be withdrawn.
5. Section 1.2.2 – the rest of this section seeks to develop evidence (not about the assertions regarding consultation that precede it) but about the success or lack of it for containment. I note that the evidence is restricted to Victoria and that there is no indication as to whether it is representative – i.e., how many councils have such measures and what is the experience of each? Without that context, readers don’t know whether the evidence is cherry-picked or if outcomes vary locally. The ACT report I’ve attached has a simple conclusion – cat containment is working, with 25% of citizens outside cat containment areas reporting a cat-related problem compared to 5% of citizens in cat containment areas. A much better job needs to be done in presenting the evidence for assertions made.
6. Lines 166-168 – I don’t think your two cited references support the assertion that areas in the USA have retracted cat confinement laws. The articles referenced deal specifically with animals being leashed when in public.
7. Line 175 – per capita, not per capital.
8. Lines 182-189 – I was unconvinced by this paragraph. Firstly, what support do you have for the assertion that people assume that introducing a law will be followed by universal compliance? If that were true, penalties wouldn’t be stated alongside laws. To take a different analogy, should laws regarding animal cruelty be repealed because some people are still being cruel to animals and considerable resources are invested when striving for compliance? A better approach is to present management options and give reasoned approaches for choosing one over another.
9. Lines 250-252 – You need to explain what is meant by One Welfare, and why it is relevant to the discussion. My feeling is that to build an argument on One Welfare, you’ll need some data on the welfare of the cats within the program – vaccinations, worming, accidents, veterinary care if needed and so on. The work of Seo et al. (see below) suggests that shelter admissions may be reduced although core animal welfare matters remain unaddressed.
10. Lines 252-262 – Engagement of Stakeholders, not Engagement of Sstakeholders
11. Lines 480-493 – This information is critical to the paper and should be upfront. Given the evidence you have for the success of the program, I am perplexed as to why the council has returned to a more expensive and less effective program.
Reference
Seo A, Ueda Y, Tanida H. Health status of 'community cats' living in the tourist area of the old town in Onomichi City, Japan. Journal of Applied Animal Welfare Science. 2022;25(4):338-54. doi: 10.1080/10888705.2021.1874952.
Seo A, Ueda Y, Tanida H. Population dynamics of community cats living in a tourist area of Onomichi City, Japan, before and after the Trap-Test-Vaccinate-Alter-Return-Monitor Event. Journal of Applied Animal Welfare Science. 2023;26(2):153-67. doi: 10.1080/10888705.2021.1901226.

Minor corrections noted in the comments to authors.
Author Response
"Please see attachment"

Round 2
Reviewer 1 Report
Comments and Suggestions for Authors
A few comments to be sure that the intended meaning is as clear as possible.
Line 78-81 and following new text: Because the manuscript then rapidly moves to the penalties for owners of cats not in compliance with laws, please add a line in this section about the importance of semi-owners becoming owners and then NOT being punished under the law for the cat being roaming or due to limit laws, etc.. This is related to the statements on lines 314-6…if semi-owners become owners, then they are subject to all these issues---it is a little contradictory. I believe what is being recommended is 1. Semi-owners become owners (for everyone’s welfare) and 2. Owners should not be liable for things they can’t afford or manage. This is clear in line 431.
Lines 476-8: This feels a bit off hand or throw away and potentially up for a challenge given the recommendation for TNR. Perhaps make it clear that the differences are also about who is paying for or performing the trapping: the government or TNR groups or semi-owners. It is practical when the labor is free and s/n is low cost or fully subsidized.
Section 3.6: somewhere it is important to note that RTF alone, unless a lot of cats are coming from the same location, may not have the impact of TNR where many more cats in a given location are sterilized. An occasional RTF cat here or there is unlikely to show the changes reported on lines 498-99 because more than RTF was performed there. That connection could be made clearer so that the reader doesn’t need to review the reference to know that.
Author Response
"Please see the attachment"

Reviewer 2 Report
Comments and Suggestions for Authors
Animals-3159982 Urban cat management – critical success factors
The authors have completed a thorough revision of this paper. I found that it fell into two halves: (i) an evaluation of the Banyule initiative, which is interesting, informative and worth sharing, and (ii) a broader contextualization of what works and what doesn’t in urban cat management. Part (ii) is problematic and detracts from the good work in (i). My suggestion is for a further round of revision that removes the problematic part (ii), and focuses on the value of part (i).
(i) The Banyule initiative
This is by far the stronger of the two main components, mainly because of the sound, quantitative data presented. There remain, though, some significant problems that require addressing, specifically:
Please give the sources or the data underpinning all claims and ensure that cited references support the claims made. For example:
Lines 489-493: ‘There is also a reduced risk of disease transmission among pets, wildlife, and humans. For pet cats, prevalence of feline immunodeficiency virus (FIV), and cat-fight abscesses and cellulitis are all diminished with fewer free-roaming unsterilized cats, while for wildlife, the risk of toxoplasmosis is decreased[69].’ What data do you have to show that FIV has declined in Banyule? And why should the sterilization status of a free-roaming cat influence its likelihood to transmit toxoplasmosis? Your cited source [69], Westmann, is explicit about how to reduce FIV risk: keep cats indoors. I didn’t think that containment was a major part of your program.
Points to consider in improving the presentation as it stands are listed below.
Lines 76-80: if cats acquired by semi-owners don’t come with guidance on veterinary care and semi-owners are unlikely to be able to afford care, how will they receive information and how will they be able to act on it?
Are there plans to follow up on cats where semi-owners take on full ownership? What are the short, medium and long-term outcomes for these cats?
(ii) Broader contextualization
This is the weaker component of the paper. I detail some of the problems below, grouping comments under the sub-headings of the paper. Fixing the problems likely requires a separate paper, so I recommend removing this section from the manuscript.
Section 2.2 Role of Animal Management Officers (AMOs) in Urban Cat Management
This section contains many unreferenced assertions. Indeed, there are no references for the many statements in lines 140-154. Sources are needed so that these statements can be checked.
Section 2.3 Mandated containment
Lines 198-200 ‘In the USA, it has been recognized by many welfare agencies and local government areas that cat confinement laws are unenforceable in the absence of an identifiable owner, and most have been rescinded.’ The cited references are either general comments on cats and the law in the USA, or reports on a single, specific ordinance being repealed. Not one is peer reviewed. They do not support the statement that ‘most have been rescinded.’ For that to be true, you would need a history of all such ordinances in the USA including when they were implemented and if they are still in force.
There is no consideration of the carefully considered ACT approach to containment, which includes ‘… an educative approach to compliance measures to allow time for the community to adjust to the new requirements’ and ‘The grandfathering approach for existing pet cats will support a fair and gradual transition which supports the welfare for these animals.’ https://www.cmtedd.act.gov.au/open_government/inform/act_government_media_releases/act-transport-canberra-and-city-services-media-releases/2022/new-laws-for-cat-containment-and-cat-registration. Punitive enforcement and compelling all owners to contain or give up their cats need not be part of mandatory containment.
There is also no consideration of the finding in the Micromex report (which I provided on the first review), that 95% of surveyed residents in Forde and Bonner (where cat containment is required) believed that contained cats ‘are less likely to be a nuisance to the community’ compared to 54% of surveyed residents elsewhere in Canberra.
Lines 120-123: ‘… approximately 32 out of the 79 Victorian 120 local government areas have local by-laws (ordinances) which require cats are to be contained to the owner’s property, either dusk to dawn or 24/7, and many others are considering introducing these curfews.’ If you wanted to explore the success or otherwise of containment in Victoria, you’d need to follow up these areas for reasons for implementation, success criteria etc. At the moment, the paper simply doesn’t present these data.
Section 2.4 Mandated sterilization
Lines 244-249: ‘The ACT enacted mandatory sterilization for cats in 2001. In 2000, cat intake into the ACT RSPCA shelter was 7.69 cats/1000 residents and by 2006 cat intake was 8.02 cats/1000 residents [45]. This suggests the sterilization mandate made no positive impact on shelter intake of cats, likely because legislators failed to recognize the underlying causes, which are cost and accessiblity barriers for low-income cat owners, and that most shelter admissions are unidentified stray or semi-owned cats[16].’
Comprehensive data on the number of cats admitted to RSPCA shelters in the ACT are available at https://www.rspca.org.au/about/annual-statistics/. The archived data show a decline in the number of cats admitted to ACT RSPCA shelters from 2560 in 1999/2000 to 1501 in 2021/2022.
Year |
Total cats |
Euthanased |
% euthanased |
2021_2022 |
1501 |
133 |
8.9 |
2020_2021 |
1766 |
259 |
14.7 |
2019_2020 |
1549 |
270 |
17.4 |
2018_2019 |
1775 |
318 |
17.9 |
2017_2018 |
1773 |
323 |
18.2 |
2016_2017 |
2315 |
465 |
20.1 |
2015_2016 |
2483 |
520 |
20.9 |
2014_2015 |
2314 |
607 |
26.2 |
2013_2014 |
2265 |
640 |
28.3 |
2012_2013 |
2343 |
776 |
33.1 |
2011_2012 |
2400 |
822 |
34.3 |
2010_2011 |
2865 |
1252 |
43.7 |
2009_2010 |
2748 |
1133 |
41.2 |
2008_2009 |
2654 |
1175 |
44.3 |
2007_2008 |
2929 |
1355 |
46.3 |
2006_2007 |
2538 |
1130 |
44.5 |
2005_2006 |
2598 |
1040 |
40.0 |
2004_2005 |
2254 |
1010 |
44.8 |
2003_2004 |
2464 |
1220 |
49.5 |
2002_2003 |
2078 |
755 |
36.3 |
2001_2002 |
2296 |
785 |
34.2 |
2000_2001 |
2465 |
940 |
38.1 |
1999_2000 |
2560 |
1007 |
39.3 |
The number of cats admitted in 2021/2022 was only 58% of those admitted in 1999/2000. A Mann-Kendall non-parametric test for trends shows a highly significant decline in admissions (p = 0.0096). These figures do not consider changes in the human population – if population growth was taken into account and data given as admissions/capita, the effect would likely be even more emphatic. If no decline means that mandatory sterilization is ineffective, then what must be concluded from a significant decline?
Section 2.5 Mandated registration
Lines 272-275: While it is true that state-wide legislation requiring registration of pet cats was removed in Queensland in 2013, it is important to acknowledge that registration is still required by some local municipalities (https://www.qld.gov.au/families/government/pets/pet-laws#registration). Furthermore, if mandatory registration is ineffective, how do you explain the situation in NSW where state-wide registration is in force (https://www.cityofsydney.nsw.gov.au/pet-animal-services/register-your-cat) and, according to Chua et al. 2023, (your reference [4]) NSW had the lowest shelter admissions/capita of any state in 2018/2019?
Comments on the Quality of English Language
No comments.
Author Response
The authors would like to thank the reviewers' patience and suggestions for our manuscript. We appreciate this has taken longer than anticipated but believe the process has been beneficial.
"Please see attachment"

Round 3
Reviewer 2 Report
Comments and Suggestions for Authors
Manuscript ID: animals-3159982 Cotterell et al. Rethinking urban cat management: Limitations and unintended consequences of traditional cat management
The authors have addressed a global problem which, to date, seems to defy a single, effective solution. The paper is well-structured, with each of the major headings covering an important topic and sub-headings logically grouped to break discussion into concise management issues. Those positives aside, there are major problems with the use of data and the research literature that undermine confidence in the arguments about what does and does not work in urban cat management. My concerns are substantial so, to be concise, I limit my comments to two sections of the manuscript to illustrate the types of problems that need to be overcome before the paper could be a serious candidate for publication. I do not imply, though, that sections not covered are free of similar problems.
Mandated containment
Minor points
1. The source of the RSPCA data should be referenced. I assume it is the annual statistics page https://www.rspca.org.au/about/annual-statistics/
2. Line 275 refers to 2010-12, which does not exist on the RSPCA site. Is the correct year 2010-2011, 2011-2012 or something else? I assume 2010-2011, which seems to match some of my recalculations of percentages presented (see below).
Major points
1. The argument in using the RSPCA data is that cat admissions to the RSPCA shelters in the ACT (which has mandatory containment) declined less over the period specified than those in the RSPCA shelters in NSW (which has no mandatory containment) and less than the national rate of decline over the same period, ‘suggesting that containment is not effective in reducing free-roaming cats’ (lines 279-280). Points 2-5 below explain the problems with this argument.
2. I can replicate the percentage declines of 39% for the ACT and 62% for NSW, but not the 73% nationally – I get 48%. One of us is wrong, so it needs checking.
3. For the logic of this argument to work, one would expect that all states and territories without mandatory containment would have lower rates of admissions to RSPCA shelters than those that have mandatory containment. This doesn’t seem to be true. In SA, for example, where I don’t think containment is mandated in state legislation, admissions to RSPCA shelters increased over this period. Give the data for all six states and the two territories and the legislative situation in each and then see if there is a pattern. I don’t think there is one.
4. Furthermore, a myriad of variables contribute to admissions to shelters, so even if there was a pattern it would be problematic to attribute cause and effect. Variables include, but are not limited to: whether the shelters cover the whole jurisdiction (as is claimed for the ACT) or only part, other legislation (e.g., mandatory registration, sterilization), whether programs are in place at the level of local municipalities if not statewide, what dates programs/laws were implemented, and any programs, local or jurisdiction-wide, that may influence ownership or surrender of cats. Overall, the argument on shelter admissions is not convincing.
5. Finally, the argument on shelter admissions on a statewide basis is applied inconsistently in the paper. In the containment section, the falls in NSW admission relative to the ACT are used to argue that containment isn’t working. Yet when considering mandatory registration later in the paper, NSW is singled out for its registration and breeder permit fees creating significant barriers to cat ownership. Under the logic used previously, doesn’t the decline in NSW shelter admissions show that this is actually working to reduce cat numbers or surrenders?
6. Lines 283-299 – it’s my understanding that when the ACT enacted jusrisdiction-wide cat containment, it applied to cats born on or after 1 July (unless the cat lived in one of the 17 previously designated cat containment areas) https://www.cmtedd.act.gov.au/open_government/inform/act_government_media_releases/act-transport-canberra-and-city-services-media-releases/2022/new-laws-for-cat-containment-and-cat-registration. This prevented cases of financial distress or hardship for existing owners who could not afford to contain their cats. In short, the very cases complained of could be prevented with this simple step.
7. Lines 299-306 – I was unconvinced by the argument that because a contained cat can escape, contaiment is impractical. Dogs and cage birds escape too, so by this logic there should be no legal requirement to contain them.
TNR and RTF
Major points
1. The literature in this section is highly selective and doesn’t consider differing views or evidence. Examples follow.
2. While shelter intake can be reduced by local TNR programs, this does not necessarily equate to good welfare outcomes for the cats. As two examples among many I could provide, these papers show evidence of poor welfare outcomes for cats in TNR programs:
Seo A, Ueda Y, Tanida H. 2021. Population dynamics of community cats Living in a tourist area of Onomichi City, Japan, before and after the Trap-Test-Vaccinate-Alter-Return-Monitor Event. Journal of Applied Animal Welfare Science.
Seo A, Ueda Y, Tanida H. 2022. Health status of 'community cats' living in the tourist area of the old town in Onomichi City, Japan. Journal of Applied Animal Welfare Science 25:338-354.
The paper needs to consider when and why some programs are successful, whereas others end in poor cat welfare outcomes.
3. Lines 514-515 – To dismiss concerns about predation by free-roaming cats in urban areas as overblown and to attribute problems to habitat destruction is to dismiss a comprehensive and nuanced literature in a sentence. I could cite many data-based examples to the contrary, including these three:
Dufty AC. 1994. Population demography of the eastern barred bandicoot (Perameles gunnii) at Hamilton, Victoria. Wildlife Research 21:445-457.
Greenwell CN, Calver MC, Loneragan NR. 2019. Cat gets its tern: A case study of predation on a threatened coastal seabird. Animals 9.
van Heezik Y, Smyth A, Adams A, Gordon J. 2010. Do domestic cats impose an unsustainable harvest on urban bird populations? Biological Conservation 143:121-130.
4. Lines 515-516 – While it is valid to consider the mental health of those confronted with the problems of managing free-roaming urban cats, the people who care about animals include those who deal with the wildlife casualties of these cats, for example:
Gårdebäck A, Joäng M, Andersson M. 2024. Common Causes for Veterinary Visits among Australian Wildlife. Animals 14.
Gartrell BD, Jolly M, Tissink K, Argilla LS, Esam F. 2023. A retrospective study of native wild birds and reptiles admitted to three New Zealand wildlife hospitals due to predation by cats. New Zealand Veterinary Journal 71:86-91.
Grace Demezas K, Douglas Robinson W. 2021. Characterizing the influence of domestic cats on birds with wildlife rehabilitation center data. Diversity 13.
Comments on the Quality of English LanguageWith a few small edits, the English expression of the paper is fine.
Author Response
As per Academic Editors instruction- Please amend the manuscript in line with the comments/minor revisions from reviewer 1. We have discussed the further feedback from reviewer 2 at Animals and decided for academic reasons that we will publish the manuscript once the minor revisions requested by reviewer 1 have been incorporated.